

# Some metaheuristic algorithms for solving multiple cross-functional team selection problems

Son Tung Ngo[1,2], Jafreezal Jaafar[1], Aziz Abdul Izzatdin[1], Giang Truong Tong[2] and Anh Ngoc Bui[2]

[1] Department of Computer and Information Sciences, Universiti Teknologi PETRONAS, Seri Iskandar, Perak, Malaysia
[2] Information and Communication Department, FPT University, Hà Noi, Vietnam

## ABSTRACT

We can find solutions to the team selection problem in many different areas. The problem solver needs to scan across a large array of available solutions during their search. This problem belongs to a class of combinatorial and NP-Hard problems that requires an efficient search algorithm to maintain the quality of solutions and a reasonable execution time. The team selection problem has become more complicated in order to achieve multiple goals in its decision-making process. This study introduces a multiple cross-functional team (CFT) selection model with different skill requirements for candidates who meet the maximum required skills in both deep and wide aspects. We introduced a method that combines a compromise programming (CP) approach and metaheuristic algorithms, including the genetic algorithm (GA) and ant colony optimization (ACO), to solve the proposed optimization problem. We compared the developed algorithms with the MIQP-CPLEX solver on 500 programming contestants with 37 skills and several randomized distribution datasets. Our experimental results show that the proposed algorithms outperformed CPLEX across several assessment aspects, including solution quality and execution time. The developed method also demonstrated the effectiveness of the multi-criteria decision-making process when compared with the multi-objective evolutionary algorithm (MOEA).

## INTRODUCTION

### Background

Cross-functional team (CFT) selection (*Feng et al., 2010*) is a major area of interest in management, operational research, and other fields. Selecting the right teams brings success to an organization. We can define a CFT as a group of suitable candidates with excellent personal skills who can collaborate and support each other in their work. This team's skills are multidisciplinary and span many different fields. This article is a follow-up to a study from 2020 (*Ngo et al., 2020*) that looked to develop a methodology for selecting

Corresponding author
Son Tung Ngo, sonnt69@fe.edu.vn

CFTs from available candidates. The team selection problem belongs to the class of NP-Hard and combinatorial optimization problems [1...5]. In a single team selection problem, researchers can represent the solutions as $X = \{x_i | x_i = (0, 1), i = 1 \ldots K\}$ where $K$ stands for the number of candidates, $x_i = 1$ means that selected team contains the student $i$th, and $x_i = 0$ otherwise. To select a team with $h$ members from $C$ candidates, the number of available solutions is up to $\binom{h}{K}$ (*Ngo et al., 2020*).

In practice, the number of selected groups is usually more than one and correspond to different tasks. An increase in the number of candidates, team size, or the number of formulated teams significantly increases the search space. The number of available solutions in the search space for a $G$ teams selection problem is:

$$\binom{h_1}{K}\binom{h_2}{K-h_1}\binom{h_3}{K-h_1-h_2}\cdots\binom{h_G}{K-\sum_{i=0}^{G-1}h_i} = \prod_{g=1}^{G}\binom{h_g}{K-\sum_{i=0}^{g-1}h_i}$$

where $h_g$ denotes the team size of the group $g$th and $h_0 = 0$.

The solution is represented as a graph $\mathfrak{H} = (V, E)$, where V represents the set of $C$ candidates and $G$ groups. Each existing edge in E illustrates the group assignment of the corresponding candidate. Figure 1 shows an example of the CFT selection problem.

### Related research

The optimization of modern team selection aims to achieve many goals while also following business requirements. Therefore, there are the difficulties of both the classical problem and multi-objective optimization problem (MOP). The desired goals are often improved performance, cost, or benefits. The selected members/ teams must cooperate to solve common problems and achieve a specific purpose. Employers need to maximize profits when selecting team members from the available candidates (*Wang & Zhang, 2015*). Edmondson & Harvey also emphasize that team members tend to discuss shared instead of unique knowledge, even if their individual experience is vital to their group's efforts (*Edmondson & Harvey, 2018*). Table 1 shows that most previous studies aimed to select one group. Selecting multiple teams forces the resolution of repeated issues by eliminating the selected candidates. This leads to the unfair treatment of the groups that are chosen later.

MOP techniques are categorized into no-preference, preference, posteriority, and interactive (*Ojstersek, Brezocnik & Buchmeister, 2020*). Table 2 displays the comparison of these techniques. Each method has its advantages and disadvantages. There is a need for a suitable strategy when the decision-maker may not have predefined information for the trade-off between multiple objectives (*Gunantara, 2018*), but still needs to maximize the automation of the decision-making process.

Mathematical methods and metaheuristics are two resolution techniques used for MOP and combinatorial optimization. While mathematical methods can always find the global solution (*Chand, Singh & Ray, 2018*), metaheuristics may not have that same guarantee but are more suited to large-scale applications in practice. Of the several different metaheuristic techniques, many researchers have used evolutionary algorithms

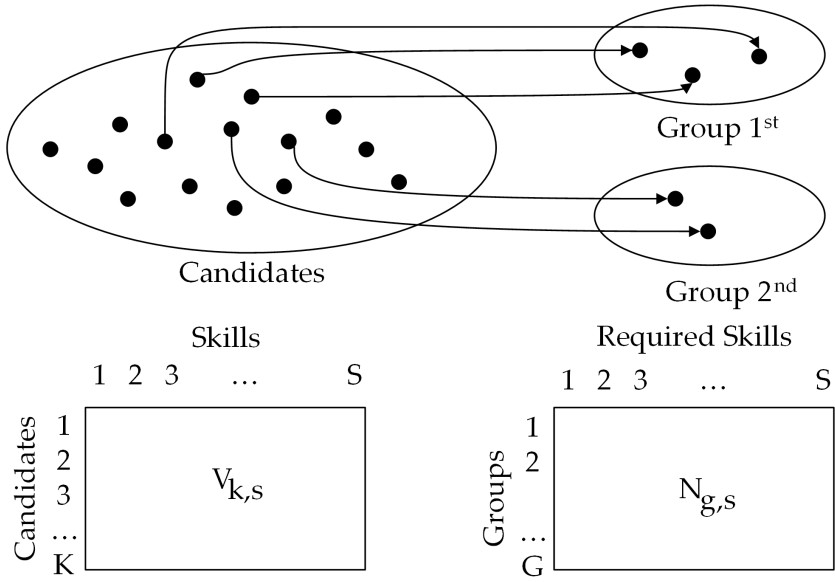

**Figure 1** Example of multi-team selection problem with $G = 2$, $h_1 = 2$, $h_2 = 2$ and $K = 15$.

(EA) and their version of MOP (MOEA) to approach MOP and combinatorial optimization problems, including the team selection problem. *Ahmed, Deb & Jindal (2013)* designed an NSGA-II algorithm to search for a Pareto frontier in 3D space. *Zhao & Zhang (2018)* developed some metaheuristic algorithms for team foundations. *Bello et al. (2017)* built an ACO algorithm to select a team based on the preferences of their two decision-makers. State-of-the-art EAs can solve traditional combinatorial optimization problems with high-quality solutions and reasonable execution time. The MOEA approach yields results that lie on the Pareto frontier and enables decision-makers to work with different scenarios. However, these algorithms often have high execution times for practical problems. When using EAs, designers do not have to consider assumptions about the convexity and separability between the objective function and constraints. However, they may use a costly evaluation function or look for solutions in an extensive feasible set. This limitation is particularly crucial when tackling computationally expensive tasks. Therefore, it is vital to design an EA scheme to find solutions at an acceptable execution time without affecting the quality of the solutions (*Chugh et al., 2019*).

## Contributions

This study presents a solution for selecting multiple groups from candidates that match the skills criteria—an improvement from previous research (*Ngo et al., 2020*). Various aspects of the team members' skills are set as goals to be achieved by the optimizer. We used the compromise programming (CP) approach for MOP. We designed GA and ACO schemes for the proposed model. To evaluate the efficiency of the algorithms, we compared them with CPLEX's MIQP-solver. Our study suggests that there is a new variant in team selection problems. Our results benefit researchers in the field of management, as well as in empirical research on combinatorial search. This research also contributes to our

**Table 1   Summarization of some previous research in team selection.**

| Research | Objectives | Techniques | Pros and Cons |
|---|---|---|---|
| *Wang & Zhang (2015)* | Maximum benefit of the candidates | Agent-based negotiation algorithm | Single team, based on negotiation. |
| *Ahmed, Deb & Jindal (2013)* | Maximize of batting performance, bowling performance, fielding performance | NGSA-2 | Single team selection, CPU time maybe high for larger dataset. |
| *Chand, Singh & Ray (2018)* | Five objectives including batting, bowling, cost, star power, and fielding | $\varepsilon$-constraint approach, integer linear programming | Single team |
| *Pérez-Toledano et al. (2019)* | Cost, performance, valuation of player | NGSA-2 | Single team selection, CPU time maybe high for larger dataset. |
| *Ngo et al. (2020)* | Deep and Wide on the available skills. | Genetic Algorithm, DCA and ILP | CPU time maybe high for larger dataset, only appliable for single team selection |
| *Pantuso (2017)* | Maximizing the expected net present value of the team | Cplex | Single objective, multiple team |
| *Feng et al. (2010)* | Individual and collaborative performance | NGSA-2 | CPU time maybe high for larger dataset |
| *Fan et al. (2009)* | Individual and collaborative information | NGSA-2 | CPU time maybe high for larger dataset |
| *Su, Yang & Zhang (2016)* | Individual knowledge competence, knowledge complementarity, collaboration performance | Genetic Algorithm | High performance, single team selection |
| *Sharp et al. (2011)* | Maximize rate | Genetic Algorithm | Model lacks collaboration between players. |

**Table 2   Different approaches to MOP.**

| Criteria | No-Preference | Preference | Posteriority | Interactive |
|---|---|---|---|---|
| Number of Solutions: | Final solution | Final solution | Subset of the Pareto optimal solutions | Subset of the Pareto optimal solutions |
| Higher level information to indicate final solution. | No | Yes | Yes | Yes (At every iteration) |
| Solver | | EA/Mathematical Programming | MOEA/ Mathematical Programming | Algorithms use psychological convergence |
| Applicable Different Decision-Making Scenarios | No | Multiple executions with different parameters | Single execution | Decision-makers participated in the whole process |

proposed methodology to develop an effective method for multi-objective scheduling and planning problems based on CP (*Son et al., 2021c*). To evaluate the effectiveness of the CP-based approach, we assessed proposed algorithms using NGSA-II, a posteriority approach for MOP. We organized the rest of this report in the following sections. Sections 2 and 3 describe the proposed model and algorithm, respectively. To evaluate the proposed approach, we describe the experiments and discussion in Section 4. Finally, Section 5 offers a conclusion.

## Proposed optimization model
### *MOP-TS model*
The following are several variables used for the model:

- $K$ is the number of candidates.
- $G$ denotes the number of groups.
- $S$ represents the number of skills in the skillset.
- $Z_g$ stands for the number of members in group $g$th $\forall g = 1, \ldots, G$.
- The decision variables $X \in \mathbb{R}^{K \times G} = x_{k,g} | x_{k,g} \in (0,1); k = 1 \ldots K, g = 1 \ldots G$ where:
- $x_{k,g} = \begin{cases} 1 & \text{if member } k\text{th selected to group } g\text{th} \\ 0 & \text{otherwise} \end{cases}$
- $N_{g,s} = \begin{cases} 1 & \text{if skill } s\text{th is required to group } g\text{th} \\ 0 & \text{otherwise} \end{cases}$ $\quad \forall s = 1, \ldots, S; g = 1, \ldots, G.$
- $V_{k,s}$ is the rating score for skill $s$th of the candidate $i$th $\forall s = 1, \ldots, S; k = 1, \ldots, K.$
- $L_{g,s}$ is the minimum required score for skill $s$th for group $g$th.

The objective functions were defined as follows:
- To select candidates who were fluent in the required skills by group:

$$\max \left( f_g^{\text{deep}}(X) = \sum_{i=1}^{K} \left( \sum_{s=1}^{S} x_{i,g} * N_{g,s} * V_{i,s} \right) \right) \quad \forall g = 1, \ldots, G.$$

- To select candidates who knew many of the required skills by group:

$$\max \left( f_g^{\text{wide}}(X) = \sum_{i=1}^{K} \sum_{s=1}^{S} \min(1, x_{i,g} * V_{i,s} * N_{g,s}) \right) \quad \forall g = 1, \ldots, G.$$

Subject to:
- No candidate could join more than one group:

$$\sum_{g=1}^{G} x_{i,g} \leq 1 \quad \forall i = 1, \ldots, K. \qquad (C1)$$

- No group was over team size:

$$\sum_{i=1}^{K} x_{i,g} = Z_g \quad \forall g = 1, \ldots, G. \qquad (C2)$$

- Selected groups must respect the minimum required skills:

$$\sum_{k=1}^{K} x_{k,g} * V_{k,s} \geq L_{g,s} \quad \forall g = 1, \ldots, G; s = 1, .., S. \qquad (C3)$$

### *Compromise programming to MOP-TS*
The idea of compromise programming (CP) (*Ringuest, 1992*) is based on not utilizing any preference information or depending on assumptions about the relevance of objectives. This approach does not strive to discover numerous Pareto solutions. Instead, the distance between a reference point and the feasible objective region is reduced to identify a single optimal solution. Many studies have used CP for their MOPs, such as examination

timetabling (*Tung Ngo et al., 2021*), teacher timetabling (*Ngo et al., 2021b*), enrollment timetabling (*Ngo et al., 2021a*), learning path recommendation (*Son et al., 2021a*), and task assignment (*Son et al., 2021b*). Several methods are used to select the preferred point (*Ngo et al., 2021c*) or normalize the distance function (*Ngo et al., 2022*). In this situation, we introduced the compromised objective function as: $\min\left(\sum_{i=1}^{L} w_i |\frac{F_i - z_i^*}{z_i^{worst} - z_i^*}|^2\right)^{1/2}$ where $L = 2 * G$ and $z_i^{worst} = \max_{x \in X} f_i(x)$.

The $z^*$ and $z^{worst}$ were pre-calculated as:

- $z^{worst} = \{d^{min} | c^{min}\}$
- $z^* = \{d^{max} | c^{max}\}$
- $F = f^{deep} | f^{wide}$.

Where:

- $d^{min} = \{d_g^{min} | d_g^{min} = \sum_{s=1}^{S} \left(\sum_{i=1}^{Z_g} N_{g,s} * P_{s,i}\right), g = 1, \ldots, G\}$
- $c^{min} = \{c_g^{min} | c_g^{min} = \sum_{s=1}^{S} \min(1, P_{s,1} * N_{g,s}), g = 1, \ldots, G\}$
- $d^{max} = \{d_g^{max} | d_g^{max} = \sum_{s=1}^{S} \left(\sum_{i=1}^{Z_g} N_{g,s} * R_{s,i}\right), g = 1, \ldots, G\}$
- $c^{max} = \left\{c_g^{max} | c_g^{max} = \sum_{s=1}^{S} \sum_{i=1}^{Z_g} \min\left(1, R_{s,1} * N_{g,s}\right), g = 1, \ldots, G\right\}$ and
- $R_s$ is sorted vector of $\left(V_{1,s}, V_{2,s} \ldots, V_{K,s}\right)$ by descending order.
- $P_s$ is sorted vector of $\left(V_{1,s}, V_{2,s} \ldots, V_{K,s}\right)$ by ascending order.

## Proposed algorithms

Researchers have designed meta-heuristic algorithms to solve combinatorial optimization and NP-Hard problems. This section describes the design of two EAs (*Katoch, Chauhan & Kumar, 2021*) used to solve the proposed model: the genetic algorithm (GA) and ant colony optimization (ACO).

### *Genetic algorithm*

The GA scheme is illustrated in Fig. 2, where the chromosome is represented like the decision variables $X$, but here we used list $p$ as List $<$ List $<$ Integer $>>$ of $G$ items. Each $p_g$ stores the indexes of selected candidates (Fig. 3). This mechanism allows for reducing the size of the original $X$.

The steps of the algorithm (Fig. 2) are described as:

- Generation of the initial population: We generated population $P$ as the set of $\pi$ indiviuals.
- Computation of the fitness function: For each $p$ in $P$ as:

$$p.\text{fitness} = \begin{cases} \left(\sum_{i=1}^{2*G} w_i \left|\frac{F_i - z_i^*}{z_i^{worst} - z_i^*}\right|^2\right) & \text{if } val(p) = 0. \\ 1 & \text{otherwise} \end{cases}$$

Where $val(p)$ returns 1 if the solution $p$ violates any constraints of $(C1), (C2)$, or $(C3)$. The function returns 0 if $p$ is valid. The violated individuals are punished by removing their effects. These solutions are potentially replaced by better quality solutions in the following genetic operations.

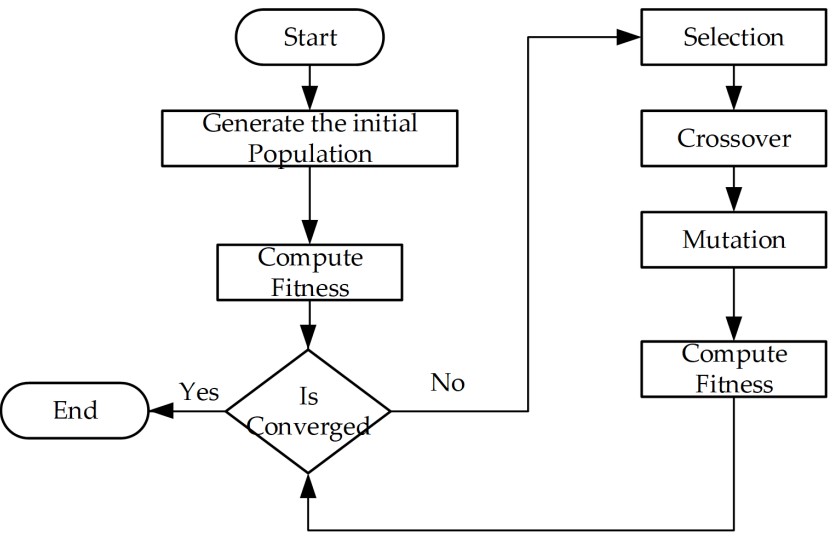

**Figure 2** Basic flow of genetic algorithm.

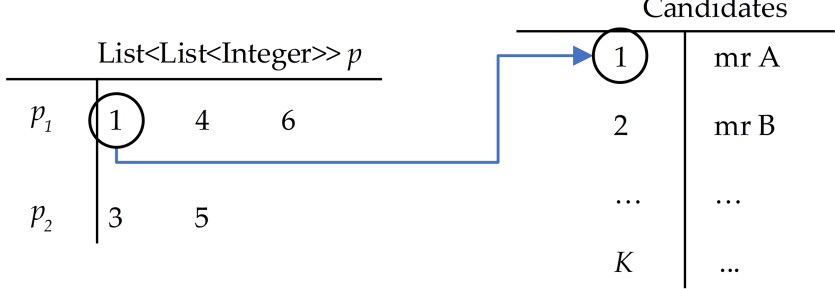

**Figure 3** Example of the solution in GA with G = 2 candidates with id {1, 2, 3} selected for team 1, candidates with id {3, 5} selected for team 2.

- Selection: We chose the selection rate of $\phi$ to keep the elite individuals in the next generation.
- Crossover: $\mu$ denotes the crossover rate where the new individual was constructed as follows:
  1. Randomly select two individuals as parents denoted by $p_1, p_2$
  2. Gather selected members from $p_1, p_2$ to list pool.
  3. Randomly select items from pool to construct the new solution for the next generation.

  Figure 4 illustrates an example of the three steps of the crossover phase.

- Mutation: We used the mutation rate $\beta$ to select individuals from the population. These individuals' genes were modified by randomly selected candidates from the list of $K$ candidates.

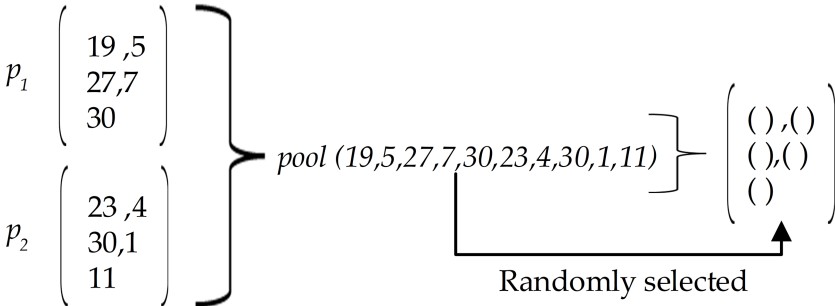

**Figure 4** An example of the crossover phase.

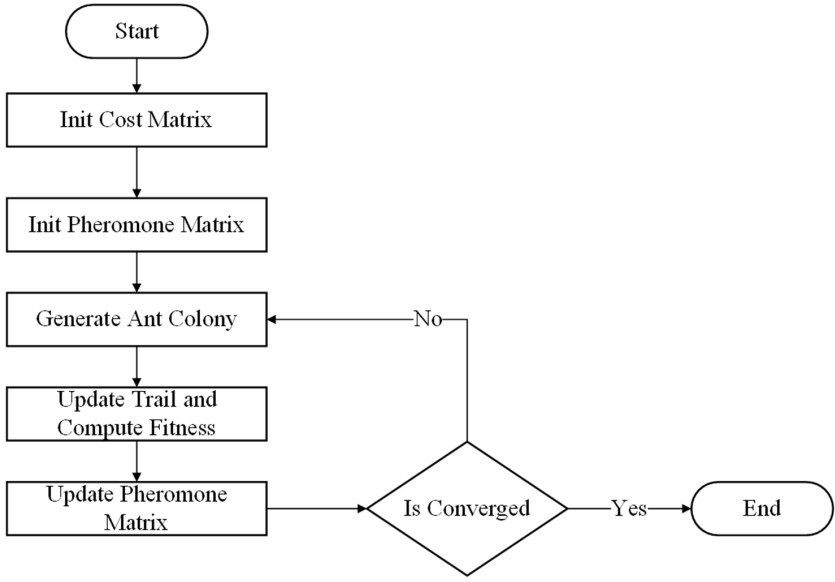

**Figure 5** Basic flow of ant colony optimization.

- We defined a condition that if after $\alpha$ generations, the system could not find any better solutions, the algorithm would stop. Otherwise, continue to the selection phase.

### ACO for multiple team selection

To design the scheme of the TS-ACO, we applied a similar data structure to represent the artificial ant (solution) as GA. Figure 5 shows the flow chart of the proposed ACO algorithm.

The details of each step of the ACO is described as follows:

- Initialization of the cost matrix: We created matrix $C \in \mathbb{R}^{K \times G}$ where $C_{k,g}$ represented the cost if candidate $k$th was chosen for group $g$th as following:

$$C_{k,g} = \sum_{z=1}^{2} w_{g*z} \left| \frac{M_{z(k,g)} - z_g^*}{z_g^{\text{worst}} - z_g^*} \right|^2.$$

Where:

$$M_1(k,g) = \sum_{s=1}^{S} (N_{g,s} * V_{k,s})$$

and

$$M_2(k,g) = \sum_{1}^{S} (\min(1, N_{g,s} * V_{k,s})).$$

- Initialization of the pheromone matrix: We created matrix $T \in \mathbb{R}^{K \times G}$ where $T_{k,g} = \frac{1}{C_{k,g}}$.
- Generation of ant colony: We created new population $P$ as the set of $\pi$ artificial ants.
- Update of trail and computation of fitness: the population $P$ was constructed as: $r_z = \text{rand}(v'_g), \forall z = 1 \ldots Z_g$ are selected as candidates for group $g$th, $\forall g = 1 \ldots G$ of the ant $p$, $\forall p \in P$, where:

  - rand represents the cumulative distribution function.
  - $v_g$ denotes vector column $g$th of matrix $T$.
  - $v'_g = \left\{ \frac{v_{g,1}}{\sum_{i=1}^{K} v_{g,k}}, \frac{v_{g,2}}{\sum_{i=1}^{K} v_{g,k}}, \ldots, \frac{v_{g,K}}{\sum_{i=1}^{K} v_{g,k}} \right\}.$

- Update of pheromone matrix: We selected the top $\Phi$ of the best solution from $P$ to list $E$, and then the $T$ was updated as following:

$$T_{p_{z,g},g} = \frac{1}{C_{p_{z,g},g}} \quad \forall p \in E, g = 1 \ldots G, z = 1 \ldots Z_g.$$

- We also defined a condition that if, after $\alpha$ generations, the best fitness value of the population did not change, the algorithm stopped. Otherwise, we returned to the step of generating a new ant colony.

### Computational complexity

Both GA and ACO operations are stochastic. It is difficult to determine the exact complexity of these algorithms, although we can calculate the cost of each iteration (Table 3). However, it is difficult to predict the number of iterations the algorithm will need to perform in advance when GA is $O(K * H * S)$ and ACO is $O(K * H * S * G)$. However, the number of iterations is not predictable. Thus, if we assign $N_{GA}$ and $N_{ACO}$ as the number of iterations for each algorithm, we can say that the complexity of GA and ACO are $O(N_{GA} * K * H * S)$ and $O(N_{ACO} * K * H * G * S)$, respectively. We predicted that the computation time of GA would be better than that of ACO. We used execution time (CPU time) to illustrate this in our experiments.

### Experiments and results
### Experimental design

To evaluate the performance of the proposed algorithms on a real-world dataset, we used the dataset of 500 programming contestants (Ngo et al., 2020). Contestants tackled programming exercises and were each tagged into different types of exercises across 37
**Table 3  Computational complexity of each iteration in GA and ACO algorithms.**

| GA | | | ACO | | |
|---|---|---|---|---|---|
| Step | Description | Complexity | Step | Description | Complexity |
| 1 | Compute Fitness | $O(G*H)$ | 1 | Compute Fitness | $O(G*H)$ |
| 2 | Generate individuals | $O(K)$ | 2 | Generate Cost matrix | $O(G*K)$ |
| 3 | Selection | $O(K)$ | 3 | Generate Pheromone matrix | $O(G*K)$ |
| 4 | Crossover | $O(K)$ | 4 | Generate Ant Colony | $O(K)$ |
| 5 | Mutation | $O(K)$ | 5 | Update Pheromone Matrix | $O(G*K)$ |

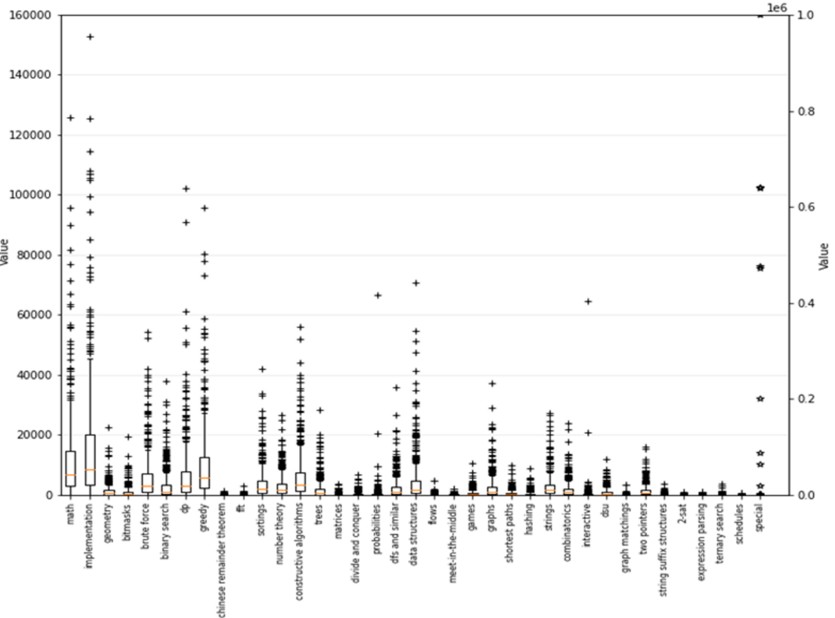

**Figure 6  Statistical numbers on 37 skills in the dataset of 500 programming contestants.**

categories. Figure 6 shows the statistical numbers corresponding to the available skills in the dataset.

We displayed the dataset with a two-dimensional space using t-SNE to find a similar probability distribution across the contestants in low-dimensional space (Fig. 7). Each point represents a contestant in the original space. The locations of the points indicate that the class difference between the contestants was very distinct. This allocation affects the search results because some members often appear in most of the solutions because they dominated in several skills. A dataset of the normal distribution can have many Pareto solutions.

The quality of the metaheuristic solution depends on factors such as the distribution of the data parameters. Since the algorithms are designed based on stochastic operations, it is not easy to evaluate the algorithmic complexity. Besides using the benchmarking dataset,

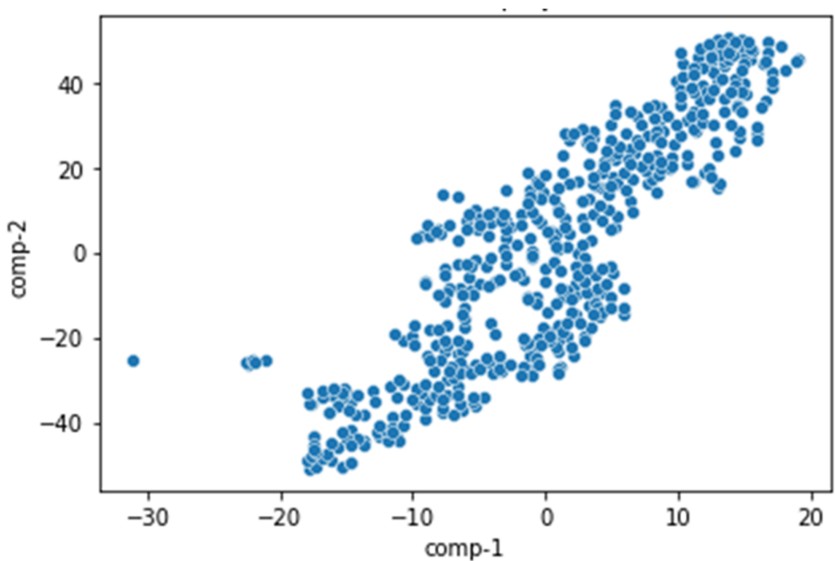

**Figure 7 The candidates are projected to two-dimensional space using t-SNE.**

**Table 4 System configurations for experiments.**

| Item | Info |
| --- | --- |
| CPU | Intel(R) Core (TM) i5-8350U CPU @ 1.70 GHz 1.90 GHz |
| RAM | Corsair Vengeance LPX 8GB |
| Programming Platform | Python 3 |
| Operating System | Window 10 |

we also generated 50 random datasets based on different distributions to compare the solution quality based on the statistical method. The 50 datasets included 300 candidates, and 30 skills were randomized based on one of six random distributions (Hypergeometric, Poisson, Exponential, Gamma, Student, and Binomial) using the Python SciPy library. We conducted the experiments on configured computers (Table 4).

Metaheuristic algorithms operate according to user customization through parameters. They significantly affect the performance of the algorithms. For example, one can bulk order search agents to increase the likelihood of finding a better-quality solution. However, this increases the execution time. We calibrated the parameter values (Table 5) used in this experiment by re-executing the algorithm several times.

## Results

The results are described in three subsections. The first section compares the designed algorithm with the designed GA by *Ngo et al. (2020)* for a single team selection problem. The second subsection shows the results of multiple team selection. To assess the performance of the compared algorithms, we considered the aspects of the quality of the optimal solutions,

**Table 5** Parameters to conduct the experiments.

| Parameter | GA | ACO |
|---|---|---|
| Population size ($\pi$) | 2* $K$ | 0.6* $K$ |
| Crossover rate ($\mu$) | 0.9 | None |
| Mutation rate ($\beta$) | 0.1 | None |
| Selection rate ($\phi$) | 0.1 | 0.1 |
| Stop condition ($\alpha$) | 40 | 40 |

**Table 6** The results of different algorithms to select a single group that requires 37 skills on the tested dataset.

| Algorithm | K | Objective value | Rate | Execution time (s) |
|---|---|---|---|---|
| GA | 500 | 659108 | 100% | 10.4 |
| ACO | 500 | 659108 | 100% | 50.7 |
| GA-1 | 500 | 659108 | 100% | 0.23 |
| MIQP-CPLEX | 500 | – | | – |
| GA | 300 | 1005773 | 100% | 11.9 |
| ACO | 300 | 1005773 | 100% | 20.6 |
| GA-1 | 300 | 1005773 | 100% | 0.23 |
| CPLEX | 300 | 1745300 | 57% | 3.42 |

processing time, and how they dealt with different decision scenarios. The final subsection compares metaheuristics and the exact algorithm on randomly generated datasets.

(1) Single Team Selection

In *Ngo et al. (2020)*, the author compared GA algorithms (called GA-1) and DCA, and their GA-1 showed superior results when selecting three members from the tested dataset. To evaluate the quality of the proposed algorithms, we compared our designs to GA-1 when solving the single team selection problem using the same objective function. The results in Table 6 show that CPLEX could not find a solution from 500 candidates on the tested machine due to an out-of-memory error. Even when the number of candidates was decreased to 300, the objective value obtained by CPLEX was insufficient. GA-1 was designed to solve the single team selection problem. Meanwhile, the proposed GA and ACO aim to find multiple-team selections, but they can find a quality solution as GA-1 for single team selection.

(2) Multiple Team Selection

As mentioned in Section 2.2, the original objective functions were transformed into a distance function from the actual solution point to the ideal point. This requires the use of $z^*$ and $z^{\text{worst}}$. The values of their elements are shown in Table 7. These values were readily calculated based on the candidate achievement data. We pre-calculated them for different scales of top $K$ candidates from the tested dataset. The pair values $i \in \{1, 2\}, \{3, 4\}$ and $\{5, 6\}$ of $z_i^*$ and $z_i^{\text{worst}}$ represent the best and worst scores, respectively, of the groups 1st, 2nd, and 3rd.

**Table 7  The ideal points and worst points in different scales of the system.**

| K | $z^*$ | | | | | | $z^{worst}$ | | | | | |
|---|---|---|---|---|---|---|---|---|---|---|---|---|
| | $z_1^*$ | $z_2^*$ | $z_3^*$ | $z_4^*$ | $z_5^*$ | $z_6^*$ | $z_1^{worst}$ | $z_2^{worst}$ | $z_3^{worst}$ | $z_4^{worst}$ | $z_5^{worst}$ | $z_6^{worst}$ |
| 500 | 3597600 | 66 | 3454630 | 58 | 1689677 | 25 | 0 | 0 | 0 | 0 | 0 | 0 |
| 300 | 3073288 | 66 | 2930317 | 58 | 1329388 | 25 | 1.83 | 1.83 | 1.83 | 1.83 | 0 | 0 |
| 100 | 2887750 | 66 | 2744861 | 58 | 1165651 | 25 | 11402 | 19 | 9406 | 16 | 1762 | 5.6 |
| 50 | 2617117 | 66 | 2474233 | 58 | 1165651 | 25 | 17511 | 25.45 | 16654 | 22.45 | 3992 | 7.6 |

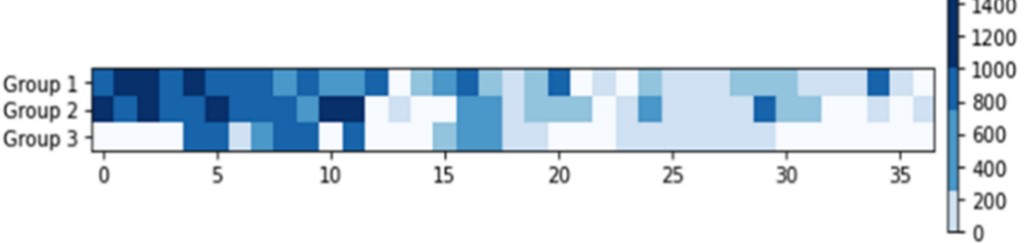

**Figure 8  The query for multiple-team's selection with G = 3.**

In this experiment, we used the query shown in Fig. 8. There were three selected groups, along with the required skills required for each group. We used the indexes of skills in arrays instead of listing their names. This query was used in the next section. The heat map illustrates the minimum required scores that needed to be archived by the selected team's skills. No required skills are displayed in white.

Figure 9 represents the results of 15 executions of each algorithm's GA, ACO, old GA, and GA using the previous search operations (*Ngo et al., 2020*) to eliminate individuals that violated the constraint. GA consistently showed a better median and the best solution when compared to old GA and ACO. In terms of time execution, old GA and GA easily outperformed ACO, and, due to the mechanism of removing solutions that violated constraints, old GA ran slower than GA. This shows that this algorithm seems to reduce the diversity of the population, therefore reducing the chances and speed of acquiring good solutions. CPLEX's fitness value and timely execution results were overwhelmed by the three algorithms above. Therefore, we concluded that our proposed scheme can adapt better than the design of *Ngo et al. (2020)* for multiple team selection situations and our designed algorithms are superior to CPLEX based on these statistical results.

Figure 10 illustrates the performances of the tested algorithms across different system sizes: 50, 100, 300, and 500 candidates. The ACO was configured to achieve similar search results as GA. Calculations and updates on cost and pheromone matrices are increasingly expensive, especially as the size of the search space expands. This setting causes the processing time of ACO to be a few times higher than that of GA, and this cost is increasing, corresponding to the candidate's size. The search operations of ACO require that every ant chooses its path by calculating the probability of each member to be selected for the groups in each iteration. Meanwhile, GA does not have to scan the whole candidate

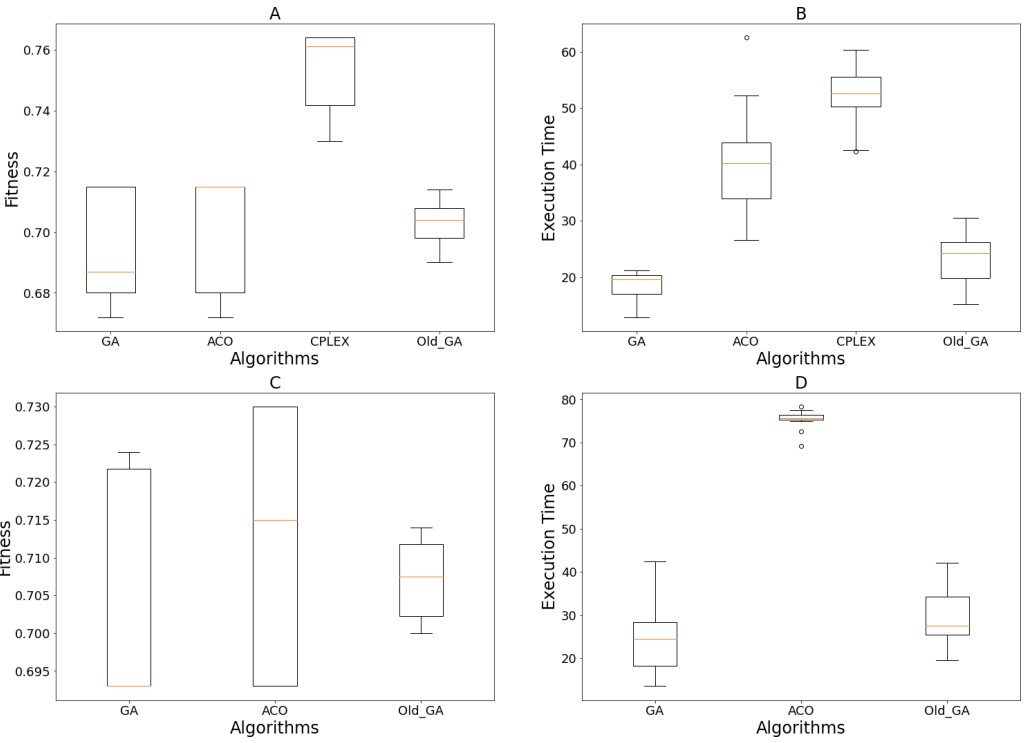

**Figure 9** **Obtained results of 15 executions of the GA, ACO, Old GA and CPLEX on different scales of the tested dataset.** (A and B) Fitness values and execution time on 300 candidates; (C and D) fitness values and execution time on 500 candidates.

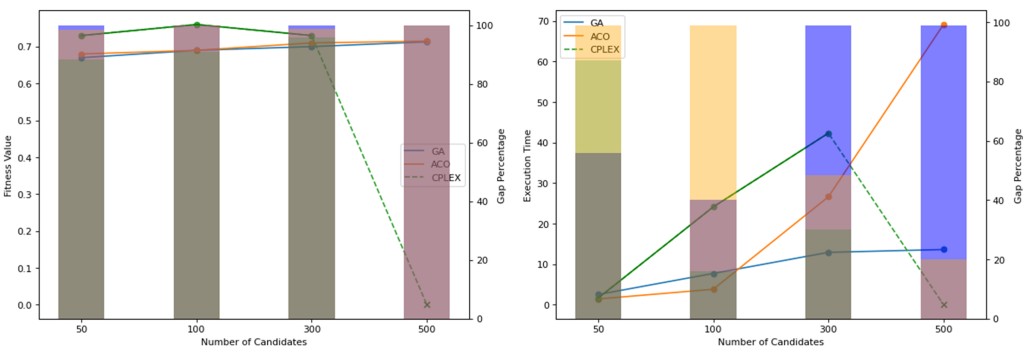

**Figure 10** **Obtained fitness values and time of computation by the algorithms on different system scales.**

for mutation or crossover. This mechanism makes ACO take longer to finish an iteration, but it allows for more search capabilities over a larger space, provided the computational resources are expanded. CPLEX is not capable of handling 500 candidates. Both CPLEX's solution quality and computation time show that it is comprehensively inferior to the proposed algorithms.

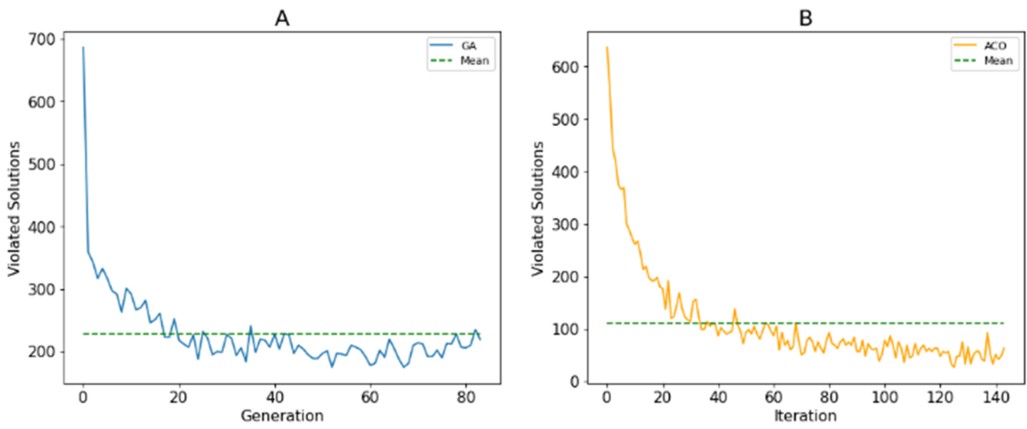

**Figure 11** The number solutions that violate the constraints generated by (A) GA; (B) ACO.

As described in Section 3A, search agents use fitness values to assess the quality of solutions. The use of stochastic operations allows agents to explore the search space. The more population diversity is ensured, the better the population's ability to discover. Unlike *Ngo et al. (2020)*, we only punish the violated resolutions to their fitness values instead of removing them or repairing them. This aims to maintain the diversity of the population. Figure 11 shows the number of violated solutions over generations for the best fitness values of the proposed algorithm with 500 candidates. This shows that the number of these solutions decreases when new generations are generated. However, the mutation/random selection process still produces a certain number of these solutions.

Fig. 12 shows the value of the fitness functions and the objective functions returned through iterations of the search processes for the best-obtained fitness values to the proposed algorithms. By observing the shape of the graphs of fitness functions produced by GA and ACO, we can see that the convergence of GA is slightly better than that of ACO. ACO's search agents need more time to complete an iteration, which leads to a costlier total processing time. GA achieved the optimal solution at the 55th iteration. Meanwhile, ACO took 101 iterations to achieve the same result. The data distribution affects the reduction of values in the distance-based fitness function. If the standard deviation is significant, the objective value has a higher impact on the fitness value calculation when solutions are projected to a specific objective function in the objective space.

(3) Different decision scenarios

Approaches to the MOP problem based on the decomposition of multi-objective functions to a ranking function have many advantages. CP is more suitable for the decision-maker who cannot indicate the preferences to trade-off the specific goals. The combination of CP and EAs allows for a seamless transition between model and algorithm design by using compromised-objective and fitness functions that are both distance-based. In contrast, it is challenging to find optimal solutions on the Pareto frontier using MOEA (*Emmerich & Deutz, 2018*). However, decision-makers use the weighted parameters to manipulate the optimizer for different decision scenarios. We executed three algorithms

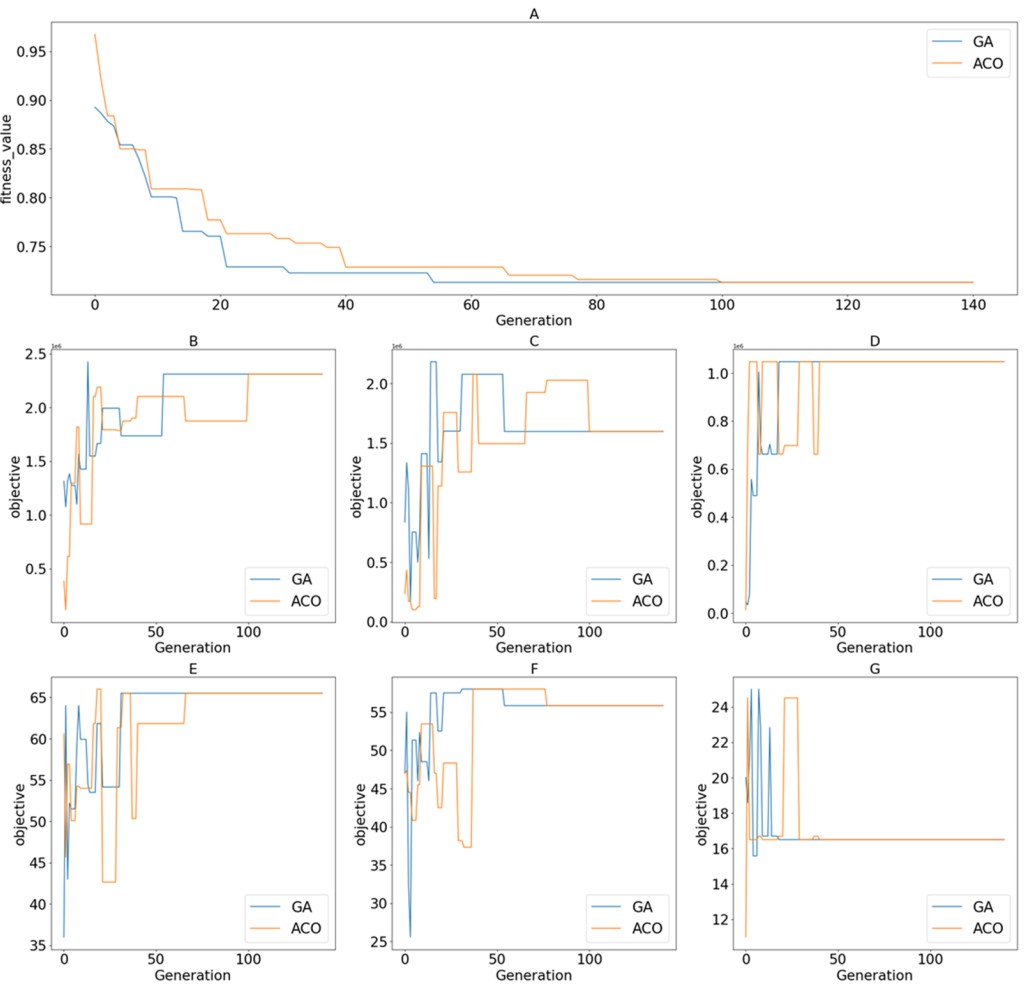

**Figure 12  Returned values of different functions over iterations/generations on the tested dataset of 500 candidates; (A) Fitness values; (B) $f_1^{\text{deep}}$; (C) $f_1^{\text{wide}}$; (D) $f_2^{\text{deep}}$; (E) $f_2^{\text{wide}}$; (F) $f_3^{\text{deep}}$; (G) $f_3^{\text{wide}}$.**

several times using 10 different sets of weight parameter values on the dataset of 300 contestants. CPLEX cannot work on the scale of 500 candidates; therefore, we only tested the proposed GA and ACO at this scale. The obtained solutions are displayed in Fig. 12 with corresponding values of the weight parameters.

The obtained solutions shown in Fig. 13 may not totally dominate each other. Therefore, it is not easy to indicate how each algorithm deals with different decision-making scenarios. To evaluate this, we measured the hypervolume *HVC* (*Ishibuchi et al., 2019*) covered by the obtained solutions. A greater *HVC* indicates that the algorithm can produce a better Pareto frontier. The hypervolume was computed as:

$$HVC = \frac{\text{volume}\left(\bigcup_{s \in S}\left(s, z^{\text{worst}}\right)\right)}{\text{volume}\left(\text{cube}(z^*, z^{\text{worst}})\right)}$$

where:

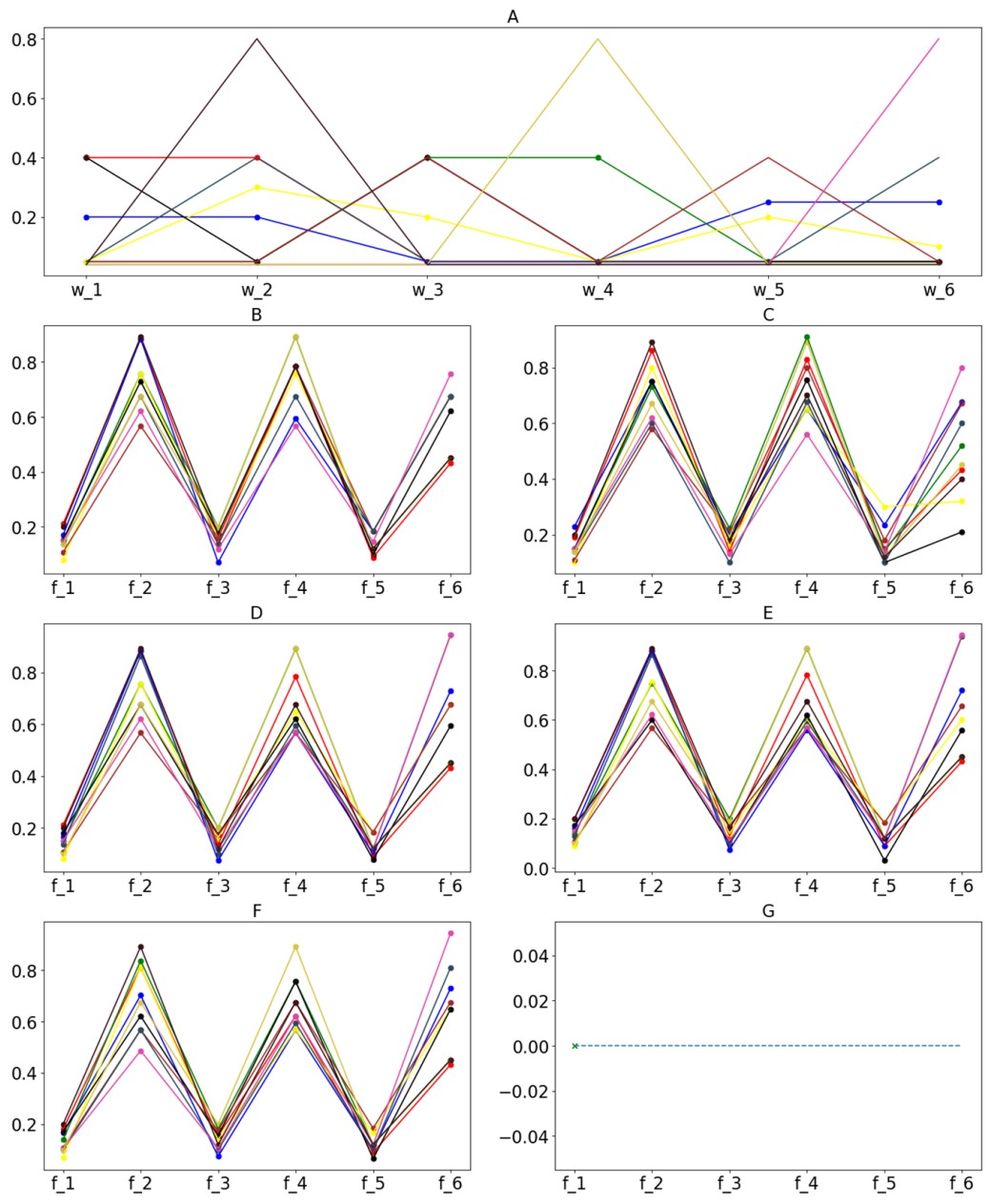

**Figure 13** **Ten optimal solutions corresponding to different weight parameters for the datasets of 300 contestants and 500 contestants.** (A) Weight parameters values; obtained solutions on dataset of 300 candidates by (B) GA, (D) ACO and (F) CPLEX; obtained solutions on data.

- $s$ denotes the solution in the Pareto solution set $S$ that is generated by the algorithm.
- cube$(a, b)$ denotes the oriented axis hypercube that is formulated by points $a$ and $b$ in the objective space.
- volume$(c)$ denotes the volume of the hypercube $c$ in the objective space.

To evaluate the capabilities of the proposed CP-based method, we used genetic operations designed to implement a version of the NSGA-2 algorithm (*Emmerich & Deutz, 2018*). The parameters used to execute the algorithm are shown in Table 8. This method has the capability to search for a Pareto front with more than 500 solutions at both scales of 300 and 500 candidates after 2,123.6 and 3,021.3 s. We used a similar cost to execute evaluated CP-based algorithms several times using different parameters. The hypervolume obtained from these solutions is displayed in Table 9. The results show that the proposed evolutionary algorithms provided superior solutions to CPLEX across different decision-making scenarios. Although the solutions generated by CP-based algorithms produced better hypervolume values than NGSA-2, it is hard to conclude that the CP-based method is better than MOEA. However, one can say that the Pareto frontier obtained by NGSA-2 may contain lower quality solutions than the proposed approach. Therefore, this approach requires significant computational overhead and is challenging to adapt to a real-world environment when the user may only need a single final solution. The user experience factor may not contribute to the effort to improve search speed.

(4) Metaheuristics *vs.* EA

CPLEX relaxes the original problem within bounds and solves the relaxation for mixed-integer linear programming (MILP) and the mixed-integer quadratic programming (MIQP). When CPLEX tries to solve a nonconvex MIQP to global optimality, it cannot provide the relaxation of the original problem that is not bounded. The proposed optimization problem is NP-Hard. The "branch and bound" may require exploring all possible permutations in the worst case (*Chopard & Tomassini, 2018*). Meanwhile, EAs (metaheuristics and general) use stochastic operations to determine possible solutions. Fortunately, the best solution can be found very early (even in the first iteration), but in the worst case, the solutions found in the first and last iteration are of the same quality. Therefore, determining the computational complexity of the two approaches from a theoretical perspective is a challenging problem and beyond the scope of this study (*Darwish, 2018*). Instead of giving theoretical calculations, we evaluated the performance of these algorithms using statistics.

Table 10 shows the results of three algorithms (GA, ACO, and CPLEX) when solving the problem using 50 generated datasets. In the 50 data sets, there were only four data sets that CPLEX found quality solutions as ACO or GA. The rest gave worse results. The execution time of CPLEX was also many times longer. While both GA and ACO solved the problem in less than 1 min, CPLEX took 3–5 min to execute. The above results show that the proposed metaheuristic algorithms are more efficient than the tested-exact method in terms of solution quality and processing time.

# CONCLUSION

This study presents an adaptive method to solve the MOP-TS problem. We introduced a MOP model for the new variant of TS problem that requires selecting multiple teams among the set of candidates. The proposed problem requires its optimizer to explore a larger space than the single TS introduced by *Ngo et al. (2020)*. The non-trivial MOPs

**Table 8   Parameters to execute and obtained results of the NGSA-2.**

| Parameter | Value |
|---|---|
| Number of Generations | 1,000 |
| Number of Populations | 1,000 |
| Selection Rate | 0.1 |
| Crossover Rate | 0.9 |
| Mutation Rate | 0.1 |

**Table 9   Hypervolume obtained by CP-based algorithms and NGSA-2.**

| Algorithm | K | Number of solutions | Hypervolume |
|---|---|---|---|
| GA | | 125 | 0.00306 |
| ACO | 300 | 82 | 0.00334 |
| CPLEX | | 40 | 0.00255 |
| NGSA-2 | | 563 | 0.00303 |
| GA | | 121 | 0.00402 |
| ACO | 500 | 79 | 0.00356 |
| CPLEX | | – | – |
| NGSA-2 | | 497 | 0.00345 |

require a trade-off between the objective functions in the decision-making process. In team selection problems, the decision-makers may not have as many suitable candidates for the teams as expected. This requires involving the higher-level information to assign preferences to each goal. We used the approach of CP to solve this problem. The solver needs to find the solution closest to the pre-assigned compromise solution instead of solving the original MOP. The proposed method integrates the designed mathematical optimization model and EAs using CP.

The compromised-objective function serves as the evaluation function of the search agents. We developed GA and ACO to solve the proposed model. To evaluate the efficiency of the proposed algorithms, we conducted many experiments to assess the algorithm's performance across different decision-making contexts. The results showed that even though the design of the proposed algorithm aimed to select multiple groups from the candidates, the quality of the solution applied to a single team selection problem, which coincided with the results of a previous study. Our method used EAs to solve the problem. We compared the proposed EAs with the exact method implemented by the CPLEX platform. Our designs showed outstanding ability when dealing with large-scale systems in terms of solution quality, execution time, and efficiency across different decisions.

The CP-based approach is beneficial when the decision-maker cannot specify priorities during the decision-making process. Even though it has a lower computation cost compared to the MOEA approach, the decision-maker can execute the algorithm multiple times and determine the values of the weighting parameters based on their experience in determining the final solution. In different decision-making strategies, the decision-makers need to use their expertise to choose the values of the parameters to motivate the search agents. The

**Table 10** Obtained results from 50 randomized generated datasets on different distribution by GA, ACO, and CPLEX.

| Dataset | Distribution | GA | | | ACO | | | CPLEX | | |
|---------|--------------|------|---------|------|------|---------|------|------|---------|------|
| | | Time | Fitness | % | Time | Fitness | % | Time | Fitness | % |
| 1 | Hypergeom | 37.6 | 0.74 | 100 | 32.7 | 0.76 | 97.36 | 253.3 | 0.82 | 90.24 |
| 2 | Hypergeom | 30.2 | 0.85 | 100 | 32.8 | 0.9 | 94.44 | 306.2 | 0.95 | 89.47 |
| 3 | Hypergeom | 39.7 | 0.75 | 100 | 37.8 | 0.76 | 98.68 | 148.3 | 0.81 | 92.59 |
| 4 | Hypergeom | 21.5 | 0.65 | 98.46 | 52.4 | 0.64 | 100 | 254.2 | 0.69 | 92.75 |
| 5 | Poisson | 36.1 | 0.72 | 95.83 | 43.9 | 0.69 | 100 | 297.3 | 0.79 | 87.34 |
| 6 | Hypergeom | 41.5 | 0.81 | 100 | 30.7 | 0.85 | 95.29 | 264.4 | 0.92 | 88.04 |
| 7 | Hypergeom | 42.5 | 0.81 | 100 | 19.9 | 0.86 | 94.18 | 236.1 | 0.93 | 87.09 |
| 8 | Hypergeom | 48.4 | 0.65 | 100 | 40.8 | 0.66 | 98.48 | 195.5 | 0.72 | 90.27 |
| 9 | Poisson | 31.4 | 0.77 | 98.70 | 19.6 | 0.76 | 100 | 264.3 | 0.82 | 92.68 |
| 10 | Poisson | 58.5 | 0.72 | 98.61 | 41.9 | 0.71 | 100 | 278.3 | 0.72 | 98.61 |
| 11 | Poisson | 59.6 | 0.72 | 97.22 | 48.4 | 0.7 | 100 | 124.8 | 0.73 | 95.89 |
| 12 | Poisson | 16.8 | 0.74 | 98.64 | 40.4 | 0.73 | 100 | 178.3 | 0.76 | 96.05 |
| 13 | Poisson | 36.5 | 0.75 | 100 | 33.4 | 0.75 | 100 | 198.6 | 0.82 | 91.46 |
| 14 | Poisson | 44.5 | 0.75 | 100 | 34.3 | 0.75 | 100 | 265.3 | 0.75 | 100 |
| 15 | Exponential | 64.4 | 0.79 | 94.93 | 41.8 | 0.75 | 100 | 179.6 | 0.82 | 91.46 |
| 16 | Exponential | 45.3 | 0.81 | 93.82 | 38.3 | 0.76 | 100 | 325.2 | 0.83 | 91.56 |
| 17 | Exponential | 44.1 | 0.81 | 93.82 | 36.6 | 0.76 | 100 | 227.3 | 0.82 | 92.68 |
| 18 | Exponential | 35.6 | 0.81 | 95.06 | 43.8 | 0.77 | 100 | 269.6 | 0.85 | 90.58 |
| 19 | Exponential | 21.8 | 0.8 | 95 | 39.8 | 0.76 | 100 | 248.5 | 0.82 | 92.68 |
| 20 | Exponential | 25.6 | 0.8 | 93.75 | 31.9 | 0.75 | 100 | 278.3 | 0.86 | 87.20 |
| 21 | Exponential | 60 | 0.82 | 95.12 | 36.2 | 0.78 | 100 | 246.3 | 0.9 | 86.66 |
| 22 | Gamma | 60 | 0.82 | 100 | 37.8 | 0.82 | 100 | 365.3 | 0.84 | 97.61 |
| 23 | Gamma | 23.5 | 0.8 | 100 | 29.5 | 0.81 | 98.76 | 212.2 | 0.82 | 97.56 |
| 24 | Gamma | 29.5 | 0.8 | 100 | 40.3 | 0.8 | 100 | 362.2 | 0.8 | 100 |
| 25 | Gamma | 52.5 | 0.82 | 98.78 | 33.6 | 0.81 | 100 | 254 | 0.82 | 98.78 |
| 26 | Gamma | 49.7 | 0.8 | 100 | 29.5 | 0.8 | 100 | 264.6 | 0.85 | 94.11 |
| 27 | Gamma | 37.8 | 0.8 | 100 | 45.6 | 0.8 | 100 | 236.3 | 0.8 | 100 |
| 28 | Gamma | 41.3 | 0.74 | 100 | 30.3 | 0.74 | 100 | 367.6 | 0.79 | 93.67 |
| 29 | Student | 44.3 | 0.71 | 92.95 | 38.7 | 0.66 | 100 | 123.2 | 0.76 | 86.84 |
| 30 | Student | 25.1 | 0.66 | 100 | 28.2 | 0.66 | 100 | 145.6 | 0.7 | 94.28 |

**Table 10** (*continued*)

| Dataset | Distribution | GA | | | ACO | | | CPLEX | | |
|---------|--------------|------|---------|-----|------|---------|-------|-------|---------|-------|
| | | Time | Fitness | % | Time | Fitness | % | Time | Fitness | % |
| 31 | Student | 23.6 | 0.66 | 100 | 22.4 | 0.66 | 100 | 198.6 | 0.66 | 100 |
| 32 | Student | 40.2 | 0.65 | 100 | 37.9 | 0.65 | 100 | 211.2 | 0.72 | 90.27 |
| 33 | Student | 24.2 | 0.65 | 100 | 30.1 | 0.65 | 100 | 214.5 | 0.73 | 89.04 |
| 34 | Student | 40.5 | 0.67 | 100 | 37.9 | 0.67 | 100 | 245.3 | 0.71 | 94.36 |
| 35 | Student | 31.9 | 0.57 | 100 | 40.5 | 0.6 | 95 | 248.6 | 0.63 | 90.47 |
| 36 | Binomial | 22.1 | 0.7 | 95.71 | 31.5 | 0.67 | 100 | 267.3 | 0.74 | 90.54 |
| 37 | Binomial | 34.5 | 0.67 | 100 | 41.4 | 0.67 | 100 | 321.2 | 0.78 | 85.89 |
| 38 | Binomial | 35.6 | 0.66 | 100 | 27.7 | 0.66 | 100 | 245.2 | 0.66 | 100 |
| 39 | Binomial | 31.4 | 0.65 | 100 | 58.4 | 0.65 | 100 | 241.2 | 0.74 | 87.83 |
| 40 | Binomial | 17.6 | 0.68 | 98.52 | 50.6 | 0.67 | 100 | 238.6 | 0.71 | 94.36 |
| 41 | Binomial | 37.8 | 0.66 | 100 | 60 | 0.66 | 100 | 234.1 | 0.7 | 94.28 |
| 42 | Binomial | 50.1 | 0.7 | 100 | 50.2 | 0.71 | 98.59 | 197.3 | 0.75 | 93.33 |
| 43 | Hypergeom | 60 | 0.75 | 100 | 30.5 | 0.78 | 96.15 | 279.2 | 0.82 | 91.46 |
| 44 | Poisson | 43.3 | 0.7 | 100 | 45.2 | 0.7 | 100 | 267.2 | 0.74 | 94.59 |
| 45 | Exponential | 36.8 | 0.61 | 100 | 40.2 | 0.63 | 96.82 | 248.3 | 0.65 | 93.84 |
| 46 | Poisson | 45.1 | 0.7 | 92.85 | 40.5 | 0.65 | 100 | 267.9 | 0.72 | 90.27 |
| 47 | Poisson | 40.7 | 0.72 | 95.83 | 49.2 | 0.69 | 100 | 249.3 | 0.76 | 90.78 |
| 48 | Poisson | 40 | 0.72 | 95.83 | 49.4 | 0.69 | 100 | 257.6 | 0.8 | 86.25 |
| 49 | Hypergeom | 32.9 | 0.75 | 100 | 26.8 | 0.78 | 96.15 | 249.3 | 0.82 | 91.46 |
| 50 | Poisson | 31.2 | 0.69 | 100 | 34.3 | 0.69 | 100 | 268.3 | 0.72 | 95.83 |

limitation of the recommended model is that it is not concerned with team communication and other soft skills, only technical skills. Therefore, our future research will focus on the development of a generic model for MOP-TS. Improving the performance of the algorithms is also one of our priorities. The complete method proposed in *Son et al. (2021c)* involves a significant amount of work. We need to evaluate the performance of the proposed method across different approaches to MOP. Problems with many objectives are also significant obstacles that require in-depth studies applying the CP-based approach. We also aim to build instructional arguments for different patterns of the SP problem.

### Funding

This research was funded by FPT University, under decision number QĐ1097/QĐ-ĐHFPT and QĐ1393/QĐ-ĐHFPT for Project HO-CPDT2021. The funders had no role in study design, data collection and analysis, decision to publish, or preparation of the manuscript.

### Grant Disclosures

The following grant information was disclosed by the authors:
FPT University: QĐ1097/QĐ-ĐHFPT and QĐ1393/QĐ-ĐHFPT for Project HO-CPDT2021.

## Competing Interests

The authors declare there are no competing interests.

## Author Contributions

- Son Tung Ngo conceived and designed the experiments, performed the experiments, analyzed the data, performed the computation work, prepared figures and/or tables, authored or reviewed drafts of the article, and approved the final draft.
- Jafreezal Jaafar conceived and designed the experiments, analyzed the data, prepared figures and/or tables, authored or reviewed drafts of the article, supervison and Directing the Theory, and approved the final draft.
- Aziz Abdul Izzatdin conceived and designed the experiments, analyzed the data, prepared figures and/or tables, authored or reviewed drafts of the article, supervison and Directing the Theory, and approved the final draft.
- Giang Truong Tong performed the experiments, performed the computation work, prepared figures and/or tables, authored or reviewed drafts of the article, and approved the final draft.
- Anh Ngoc Bui conceived and designed the experiments, performed the experiments, analyzed the data, prepared figures and/or tables, authored or reviewed drafts of the article, and approved the final draft.

## Data Availability

Data and code are available at GitHub:

https://github.com/GiangTT1909/TeamSelection?fbclid=IwAR02Corn1Q9gBMZM jUxUsuVLn_5VwcHVy82gmnfX-PMwjMQeDnWrZ83NEac.

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
