# Peer review of "Some metaheuristic algorithms for solving multiple cross-functional team selection problems"

_PeerJ Computer Science, doi:10.7717/peerj-cs.1063_

## Round 0.1 · original submission · Major Revisions

We received two consistent review reports. The reviewers pointed out the issues in the design and methodology. The paper should be revised before further consideration.

Reviewer 1 ·

Basic reporting

The highlighted Manuscript is a fair contribution for publication but requires minor improvement init

Experimental design

In the abstract, clearly highlight the computational cost section either it time of execution or memory allocation

In the proposed optimization model section, the proposed algo must highlight in tabular form

Validity of the findings

State of the art comparison is missing in it , it must be a part of this manuscript

Additional comments

In the abstract, clearly highlight the computational cost section either it time of execution or memory allocation

In the proposed optimization model section, the proposed algo must highlight in tabular form

State of the art comparison is missing in it , it must be a part of this manuscript

Add few more references of 2019,2020 an 2021
Especially in context of computational cost

Reviewer 2 ·

Basic reporting

This paper concerns Some Metaheuristic Algorithms for Multiple Cross Functional Teams Selection Problem. We designed metaheuristic algorithms to solve the proposed model, including genetic algorithm (GA) and ant colony optimization (ACO). The authors also compare the developed algorithms with the MIQP-CPLEX solver on datasets of 500 programming contestants on 37skills to evaluate their effectiveness.   
I found that the paper owns some new interests:
- method is suitable.
- the experimental results are quite clear.
- datasets are good.
- the results are quite promising.
However, the paper also has some issues that the authors need to clarify:
- The authors should explain why the existing methods in the literature cannot be applied or adjusted to solve the problem?.- The author should describe the GA and ACO in more detail. The author should show the interesting characters in two algorithms. I feel that the authors use the popular and general GA and ACO schemes though it is the first GA and ACO for the problem.- The comparison between the metaheuristic and the exact one is not realistic because they depend on two different types of algorithms. The results of exact algorithms are the optimal solution used to evaluate the efficiency of the metaheuristic.-GA1 is the authors' algorithm for a variant of the problem (Single Team Selection). The authors compare it with the GA of authors. I understand that the authors want to show that the GA algorithm in the paper can solve the variant of the problem well in the literature. The authors should explain the difference in the design of the two algorithms.- The authors should evaluate the efficiency of the proposed algorithm according to gap=(best.sol-OPT)/OPT* 100%. This formulation shows that the difference between the results of the metaheuristic with the optimal values. It is a common formulation used in the metaheuristic field.- The comparison needs to be done in a fair way. Two algorithms run on computers with the same configuration. The authors can count the fitness evaluations? The theoretical complexity is also considered.- Discussion should be added to highlight your contributions and results. 
I think the paper has some interesting points. I can be considered to publish if the authors revise well. It should be a major revision.

Experimental design

I have described it above

Validity of the findings

I have described it above

---

## Round 0.2 · Minor Revisions

A minor revision is needed. Please address the comments raised by reviewer 1.

Reviewer 1 ·

Basic reporting

The manuscript under the title " Some Metaheuristic Algorithms for Multiple Cross-Functional Teams Selection Problem" seems a mature contribution. However, some minor concerns still exist in revised version. Furthermore, grammatical ambiguities exist in the revised version.

Experimental design

1. In Introduction, the background section requires more previous contributions that will be help for readers to understand the domain. Moreover, highlighted the existing problems in the literature is not. satisfactory

2. Authors must write the proposed algorithm in standard logical form and design the flow diagram for actual execution of the proposed algorithm.

3. The Claim of least computational cost is not properly incorporated in revised version . Authors must perform experiments for supporting this claim

4. In context of computational cost, authors must define the cost factor in terms of time of execution and space consumed by executed part. Moreover, authors needs to calculate the BIG O notion if authors claim the time factor in computation cost

Validity of the findings

1.The validation of results is not satisfactory , even no single pictorial representation of the results, authors need to improve the presentation of results section.

2. The state of the art comparison still confusing , authors need to improve the state of the art comparison portion.

Additional comments

Authors need to improve the manuscript by fixing the above concerns

Reviewer 2 ·

Basic reporting

The authors revised all comments well. I think that the work can be published.

Experimental design

All experiments are conducted well.

Validity of the findings

no comment

---

## Round 0.3 · Minor Revisions

There are still some lingering comments from the reviewer. Please address them and prepare a detailed response letter. Thanks.

Reviewer 1 ·

Basic reporting

The revised manuscript under title " Some Metaheuristic Algorithms for Multiple Cross-Functional Teams Selection Problem" seems good in shape. Authors properly response all the highlighted concern . However, the computational complexity requires slightly improvement for final version of the paper. Please review the below points.

1. Authors must be incorporate the main contributions of the study in the end of introduction
2. In end of introduction section, authors highlight the rest of paper structure in organogram figure for better understanding the flow of the article

Experimental design

1. The conduction of table 2 for computational complexity requires to validate throufgh mathematical modeling.
2. Just presents Table 2 is not enough , authors need to highlight and discuss each step of BIG O notation for GA and ACO algorithm.

Validity of the findings

Authors need to present the state of the art in tabular form and clearly discuss the significance of this research.
Lastly, authors must highlight the future trends of this research that will be helpful for different research communities for proceeding of this research

Additional comments

NA

---

## Round 0.4 · Minor Revisions

The reviewers' comments have been addressed. I will recommend it for publication once the language errors have been addressed.

---

## Round 0.5 · accepted · Accept

The paper can be accepted. Congratulations.